# Gut microbiome features associate with immune checkpoint inhibitor response in individuals with non-melanoma skin cancers: an exploratory study

Yujie Zhao,[1] Jacqueline T. Ferri,[2] James R. White,[3] Megan D. Schollenberger,[2] Kim Peloza,[2] Cynthia L. Sears,[2,4,5] Evan J. Lipson,[2,5] Fyza Y. Shaikh[2,5]

**ABSTRACT** Immune checkpoint inhibitor (ICI) therapy has yielded revolutionary outcomes among some individuals with skin cancer, but a large percentage of individuals do not benefit from these treatments. The gut microbiota is hypothesized to impact ICI therapy outcomes. However, data on ICI therapy, gut microbiota, and non-melanoma skin cancers are limited. To examine the association of gut microbiota structure and function with non-melanoma skin cancer ICI outcomes, we performed 16S rRNA V1-V2 gene amplicon sequencing of 68 fecal samples collected longitudinally from individuals with basal cell carcinoma ($n = 5$), Merkel cell carcinoma ($n = 5$), or cutaneous squamous cell carcinoma (CSCC, $n = 11$), followed by tumor-dependent differential analyses of bacterial composition and fecal sample analysis by untargeted metabolomics. Across all tumor types, we identified 10 differential bacterial genera between responders (R) or non-responders (NR) to ICI therapy. Among individuals with CSCC, we identified 10 genera and 20 species that differentiated between R and NR and yielded 8 pathways enriched in NR and 12 pathways enriched in R by predicted functional pathway analyses. Untargeted fecal metabolomics to examine putative gut microbiota metabolites associated with CSCC ICI R/NR identified nine KEGG pathways associated with ICI efficacy. In summary, this exploratory study suggests gut microbiota features that are associated with ICI efficacy in individuals with non-melanoma skin cancers and highlights the need for larger studies to validate the results.

**IMPORTANCE** Prior studies examining associations between ICI efficacy and the gut microbiome have focused primarily on individuals with melanoma, for whom ICI therapy was first approved. Meanwhile, data regarding microbiome features associated with ICI responses in individuals with non-melanoma skin cancers (NMSCs) have remained limited. This initial fecal microbiota examination of individuals with NMSCs suggests that larger-scale studies to extend and validate our findings may yield predictive or prognostic biomarkers for individuals with NMSC receiving ICI with potential to provide insight to complementary, effective therapeutic interventions through microbiota modification.

**KEYWORDS** microbiome, immune checkpoint inhibitors, cutaneous squamous cell carcinoma, Merkel cell carcinoma, basal cell carcinoma

Although immune checkpoint inhibitors (ICIs) have revolutionized care for individuals with advanced non-melanoma skin cancers (NMSCs), a substantial portion of individuals experience limited benefit from these drugs. Thus, there is a growing need to identify predictive and prognostic biomarkers to guide clinical decisions and optimize individual outcomes. Among many factors, the gut microbiota appears to play an important role in influencing ICI treatment efficacy. Preclinical studies

Address correspondence to Cynthia L. Sears, csears@jhmi.edu.

Cynthia L. Sears, Evan J. Lipson, and Fyza Y. Shaikh are joint senior authors.

F.Y.S. reports research grants from Bristol Myers Squibb. J.R.W. reports equity ownership of Resphera Biosciences. C.L.S. reports research grants from Bristol Myers Squibb and Janssen and royalties from Up to Date, outside the submitted work. E.J.L. is a consultant/advisor for Bristol Myers Squibb, CareDx, Eisai, Genentech, HUYA Bioscience International, Immunocore, Instil Bio, Merck, Merck KGaA, Natera, Nektar, Novartis, OncoSec, Pfizer, Rain Therapeutics, Regeneron, Replimune, Sanofi-Aventis, Sun Pharma, and Syneos Health. Institutional research funding was provided by Bristol Myers Squibb, Haystack, Merck, Regeneron, and Sanofi. E.J.L also reports Iovance stock ownership.

See the funding table on p. 5.

using fecal microbiota transplant (FMT) with mouse models of melanoma revealed that specific species/genera enhanced ICI treatment efficacy, including *Bifidobacterium* spp., *Collinsella aerofaciens*, *Enterococcus faecium*, and Ruminococcaceae (1–3). These observations were further buttressed by clinical trials showing that FMT reversed resistance to PD-1 blockade in cutaneous melanoma (4, 5). However, little is known about the fecal gut microbiome in individuals with NMSC. In this study, we performed an exploratory analysis using 16S rRNA amplicon sequencing and untargeted metabolomics data from 68 samples collected from 21 individuals with NMSC before, during, and after ICI treatment.

Using best overall response according to RECIST v.1.1 (6), we defined individuals with progressive disease as non-responders (NR) and individuals with a partial or complete response as responders (R). Additional categories included individuals with stable disease (SD), and individuals treated with radiotherapy or organ transplant were classified as other/unevaluable. Characteristics of the individuals with NMSC studied are shown in Table S1. Figure 1A illustrates the distribution of the 68 fecal samples collected longitudinally from individuals with NMSC. Overall, 21 individuals were defined as 9 R and 6 NR, 2 with SD, and 4 as other/unevaluable.

We examined the microbial composition of all the samples using 16S rRNA amplicon sequencing with analysis using a high-resolution taxonomic assignment methodology (7) (Table S2). To investigate whether the microbiome composition changed during treatment, we separated the stool samples into three timepoint groups relative to ICI start date for each individual: samples collected within 180 days, samples collected between 180 and 360 days, and samples collected after 360 days. Since longitudinal sample collection was limited for NR after progression, longitudinal analysis using measures of alpha diversity focused on R samples. No significant differences in alpha-diversity measures were found across time points using Shannon, Simpson reciprocal, Chao1, and observed species (Fig. S1).

We then explored the microbiome composition dissimilarity of the samples among different response groups across all tumors and then tumor subtypes: basal cell carcinoma (BCC), Merkel cell carcinoma (MCC), and cutaneous squamous cell carcinoma (CSCC). Figure 1B displays the visualization by principal coordinate analysis plots showing the separations present among different clinical response groups (R, NR, stable, others) for all tumors then for each individual tumor type. To investigate whether the microbiome composition differs at the first timepoint, we consistently observed a separation by clinical response among each tumor type using the first sample for each individual collected within 180 days (Fig. S2). To identify the bacterial taxa enriched in R vs NR, taxonomic data were analyzed using linear discriminant analysis effect size (LEfSe) (8), which revealed 10 bacterial genera enriched in R across all tumors (Fig. 1C).

In order to identify tumor-specific microbiome features and due to the small sample size in the BCC and MCC cohorts, subsequent analyses focused on the CSCC cohort. Consistent with our results for all tumors, no significant longitudinal changes were identified for alpha diversity in CSCC R (Fig. S3). By LEfSe, 10 genera and 20 species were differentially enriched in CSCC R vs NR, respectively (Fig. 2A). There were no genera or species with consistently higher abundance across multiple individuals in the R group, but the genus *Bilophila* and the species *Bilophila wadsworthia* showed higher abundance for three out of four NR but low abundance in the fourth NR (Fig. 2A).

Metabolic pathways were investigated by both computation prediction from taxonomic data and direct measurement. Using taxonomic data, we used PICRUSt2 (9) used to predict functional profiles based on the MetaCyc (10) (Table S2). We identified 8 pathways enriched in R and 12 pathways enriched in NR using LEfSe (Fig. 2B). Notably, the two short-chain fatty acid (SCFA) production pathways (PWY_7254 and PWY_5022) were enriched in CSCC NR. To directly measure fecal metabolites, an untargeted metabolomics screen was performed on available samples in CSCC ($n = 6$, first sample collected within 180 days) (11) (Table S3). After filtering out the bottom 10% metabolites by peak intensity to avoid outliers, a total of 1,215 metabolites were putatively identified, with 66

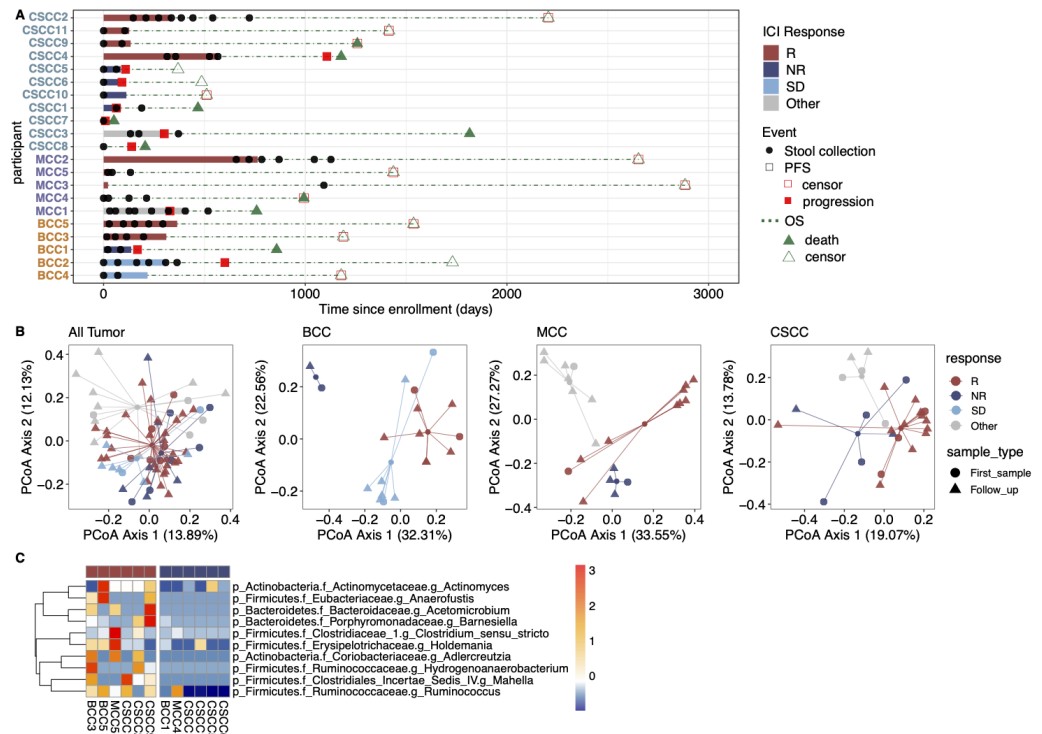

Abbreviations: IO, Immunotherapy; ICI, immune checkpoint inhibitors; BCC, Basal cell carcinoma; MCC, Merkel cell carcinoma; CSCC, Cutaneous squamous cell carcinoma; R, responder; NR, nonresponder

**FIG 1** Differential analysis by response across all tumor types. (A) Individual swimmer plots for each participant (*n* = 21), depicting each fecal sample (*n* = 68) collected at different timepoints after study enrollment for cutaneous squamous cell carcinoma (CSCC), Merkel cell carcinoma (MCC), and basal cell carcinoma (BCC). Each bar indicates the treatment length, and each black circle represents the sample collection event with respect to the IO start date. (B) Principal coordinate analysis (PCoA) of fecal samples (*n* = 68) using the Bray-Curtis distance metric across all tumor types (left), as well as within BCC (*n* = 19 samples for 5 individuals), MCC (*n* = 22 samples for 5 individuals) and CSCC (*n* = 27 samples for 11 individuals). Shape displays sample collection status, and color displays ICI responses. (C) Differential abundance analysis is performed by linear discriminant analysis effect size (*P* < 0.1, linear discriminant analysis >2) using the first fecal sample of each individual collected within 180 days (*n* = 12 individuals, each with one fecal sample). Heatmap displays relative abundance for each genus between responders (R) and non-responders (NR). Red indicates a higher abundance, and blue denotes a lower abundance. Ambiguous assignments are removed.

metabolites significantly differentiated between CSCC R and NR (47 enriched in R, 19 enriched in NR, *t*-test, cutoff of *P* < 0.1, Table S4).

To probe the function of the detected metabolites, quantitative enrichment analysis and pathway topology analysis based on Kyoto Encyclopedia of Genes and Genomes (KEGG) database (12) were performed by Metaboanalyst 5.0 (13) using differential metabolites (Table S4). Among 66 significant metabolites, 53 metabolites mapped to known KEGG pathways, and nine pathways were significantly different between CSCC R and NR (cutoff of *P* < 0.1, impact value >0; Fig. 2C). Among the significant metabolites mapped to these nine pathways, six metabolites were enriched in R and four metabolites were enriched in NR (Fig. 2D).

This study explores the gut microbiota taxonomic and putative metabolomic features associated with ICI response in individuals with NMSC. First, our study showed no significant difference in alpha diversity across various timepoints, aligning with previous research that reported microbiome stability over time in anti-PD-1-treated melanoma cohorts (14). Second, these data identified several species that might be associated with ICI efficacy. However, these species are not consistently abundant for individuals in the same group, suggesting that different organisms may serve redundant functions in the microbiota communities. Interestingly, butanoate- and acetate-producing pathways were enriched in CSCC NR, while the coexistence of *Bilophila wadsworthia* enriched in CSCC NR was previously reported to downregulate SCFA production (15). These combined results, in part contrary to the hypothesis that SCFAs promote ICI efficacy (16),

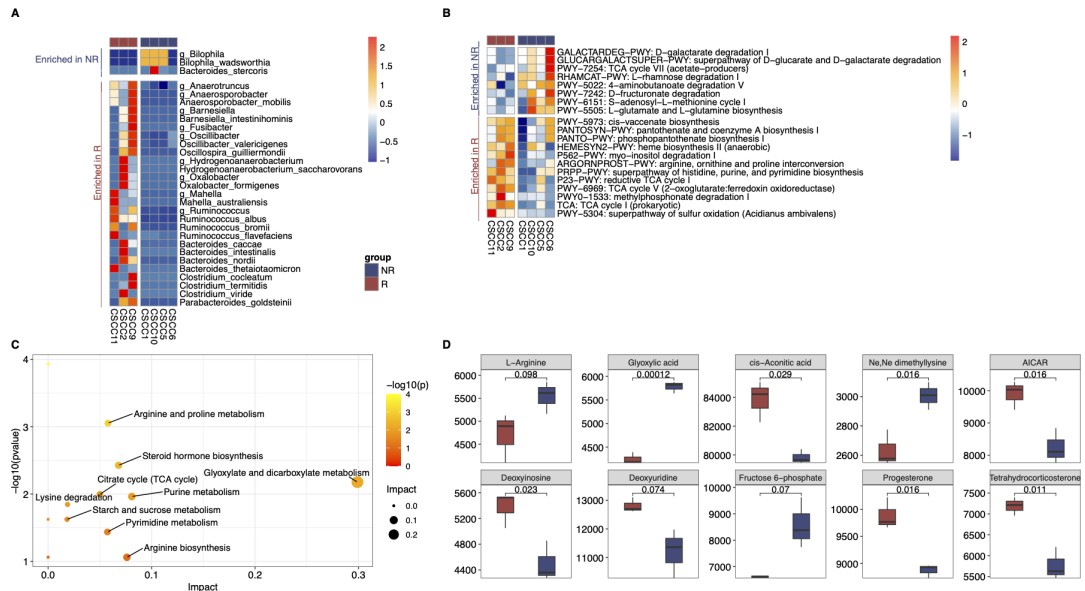

FIG 2  Differential analysis by response in cutaneous squamous cell carcinoma (CSCC). (A) Differential abundance analysis performed by linear discriminant analysis effect size (LEfSe; *P* < 0.1, LDA >2) using the first fecal sample for each individual collected within 180 days (*n* = 7 individuals, each with one fecal sample) in CSCC. Heatmap displays relative abundance for each genus between responders (R) and non-responders (NR). Red indicates a higher abundance, and blue denotes a lower abundance. (B) Differential functional pathway analysis is performed by phylogenetic investigation of communities by reconstruction of unobserved states (PICRUSt2) and displayed by LEfSe (*P* < 0.1, LDA >2) using the first samples of each individual collected within 180 days (*n* = 7, each with one fecal sample) in CSCC. Heatmap indicates relative abundance for each PICRUSt2 pathway by ICI response. Red indicates a higher abundance, and blue indicates a lower abundance. (C) Pathway enrichment analysis is performed by MetaboAnalyst 5.0 web software. Differential metabolites (*n* = 66) were identified by *t*-test. The size of the dots represents the impact value, and the color represents the *P* value. Annotated text is shown if a pathway has a *P* value of <0.1 and an impact value of >0. (D) Significant metabolites mapped to differential pathways annotated in Fig. 2C is plotted for fecal samples (*n* = 6 individuals, each with one fecal sample) in three responder (red) and three non-responders (blue). Statistics by Mann-Whitney test. The colored boxes represent the median (line inside the box) and the 25%–75% interquartile range (bottom and top edge). The upper and lower whiskers represent 95% confidence intervals.

emphasize the importance of understanding net gut microbiota community functional interactions and their impact on ICI outcomes in NMSC and other cancers. Finally, several putative metabolites of interest were identified. L-Arginine had a higher concentration associated with CSCC NR fecal samples compared to R fecal samples. In contrast, prior work has shown that a higher concentration of L-arginine in baseline plasma is associated with an improved ICI response in individuals with advanced cancer, a result also validated using a mouse model of colorectal cancer (17). Additionally, a high concentration of steroids was observed in R, including progesterone and tetrahydrocorticosterone. Two prior studies showed that progesterone can suppress the growth of melanoma *in vitro* by inhibiting pro-inflammatory cytokine interleukin-8 production (18, 19), suggesting that a similar pathway may influence NMSC growth. However, all observations herein require further validation in larger cohorts with targeted metabolomics. Despite a small sample size, this exploratory study provides initial clues that the gut microbiota may impact ICI outcomes in individuals with NMSC, laying a starting point for future translational studies to decipher key microbiome associations in this population.

## ACKNOWLEDGMENTS

The authors thank S. Glass, W. Assan, and J. Gills for their contributions to the research team. The authors also thank S. Rice, M. Leak, R. Carlson, and their respective administrative teams.

This work and C.L.S. were funded by the Bloomberg~Kimmel Institute for Cancer Immunotherapy and a research grant from Bristol Myers Squibb. F.Y.S. was supported by NIH K08CA263316 and a Damon Runyon Clinical Investigator Award. Additional support

for E.J.L. and M.D.S. was provided by The Marilyn and Michael Glosserman Fund for Basal Cell Carcinoma and Melanoma Research.

## AUTHOR AFFILIATIONS

[1]Department of Biomedical Engineering, Johns Hopkins University, Baltimore, Maryland, USA

[2]Department of Oncology, Johns Hopkins University School of Medicine, Baltimore, Maryland, USA

[3]Resphera Biosciences, Baltimore, Maryland, USA

[4]Department of Medicine, Johns Hopkins University School of Medicine, Baltimore, Maryland, USA

[5]The Bloomberg~Kimmel Institute for Cancer Immunotherapy, Baltimore, Maryland, USA

## AUTHOR ORCIDs

Cynthia L. Sears  http://orcid.org/0000-0003-4059-1661
Fyza Y. Shaikh  http://orcid.org/0000-0001-7839-8483

## FUNDING

| Funder | Grant(s) | Author(s) |
| --- | --- | --- |
| JHU \| Sidney Kimmel Comprehensive Cancer Center, Johns Hopkins University \| Bloomberg~Kimmel Institute for Cancer Immunotherapy, Sidney Kimmel Comprehensive Cancer Center, Johns Hopkins University (Bloomberg~Kimmel Institute for Cancer Immunotherapy) | | Cynthia L. Sears |
| Bristol Myers Squibb Foundation (BMSF) | | Cynthia L. Sears |
| HHS \| NIH \| National Cancer Institute (NCI) | K08CA263316 | Fyza Y. Shaikh |
| Damon Runyon Cancer Research Foundation (DRCRF) | | Fyza Y. Shaikh |
| The Marilyn and Michael Grossman Fund for Basal Cell Carcinoma and Melanoma Research | | Evan J. Lipson |

## AUTHOR CONTRIBUTIONS

Yujie Zhao, Conceptualization, Formal analysis, Methodology, Visualization, Writing – original draft, Writing – review and editing | Jacqueline T. Ferri, Methodology, Writing – review and editing | James R. White, Methodology, Writing – review and editing | Megan D. Schollenberger, Data curation, Methodology, Writing – review and editing | Kim Peloza, Data curation, Methodology, Writing – review and editing | Cynthia L. Sears, Conceptualization, Data curation, Funding acquisition, Resources, Supervision, Writing – original draft, Writing – review and editing | Evan J. Lipson, Conceptualization, Data curation, Methodology, Supervision, Writing – review and editing | Fyza Y. Shaikh, Conceptualization, Data curation, Methodology, Supervision, Writing – original draft, Writing – review and editing

## DATA AVAILABILITY

The raw sequencing data hasve been deposited under accession number PRJNA1136491. All codes utilized isare available in the GitHub repository (https://github.com/yzhao07/NMSC.git). Detailed methods and material description are available in the supplemental material.

## ETHICS APPROVAL

The study protocol and all amendments were approved by the institutional review board of Johns Hopkins University (Johns Hopkins Medicine Institutional Review Board).

## ADDITIONAL FILES

The following material is available online.

### Supplemental Material

**Table S1 (Spectrum02559-24-s0001.docx).** Participant characteristics.
**Supplemental material (Spectrum02559-24-s0002.docx).** Fig. S1 to S3; Methods.
**Supplemental table legends (Spectrum02559-24-s0003.docx).** Legends for Tables S1 to S4.
**Table S2 (Spectrum02559-24-s0004.xlsx).** Alpha diversity, clinical characteristics, and relative abundance table.
**Table S3 (Spectrum02559-24-s0005.csv).** Untargeted metabolomics data.
**Table S4 (Spectrum02559-24-s0006.csv).** Differential metabolomics in CSCC.

### Open Peer Review

**PEER REVIEW HISTORY (review-history.pdf).** An accounting of the reviewer comments and feedback.

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
