## [Reviewer comments · Microbiology Spectrum]

Microbiology Spectrum

An exploratory study: the gut microbiome associates with immune checkpoint inhibitor response in individuals with non-melanoma skin cancers

Yujie zhao, Jacqueline Ferri, James White, Megan Schollenberger, Kim Peloza, Cynthia Sears, Evan Lipson, and Fyza Shaikh

Corresponding Author(s): Cynthia Sears, Johns Hopkins University School of Medicine

Review Timeline:

Submission Date:	November 21, 2024
Editorial Decision:	November 22, 2024
Revision Received:	November 25, 2024
Accepted:	November 26, 2024

Editor: Jan Claesen

Reviewer(s): The reviewers have opted to remain anonymous.

Transaction Report:

DOI: <https://doi.org/10.1128/spectrum.02559-24>

Re: Spectrum02559-24 (An exploratory study: the gut microbiome associates with immune checkpoint inhibitor response in individuals with non-melanoma skin cancers)

Dear Dr. Fyza Y. Shaikh:

Thank you for the privilege of reviewing your work. Below you will find my comments, and instructions from the Spectrum editorial office.

Thank you for addressing the Reviewer comments from your prior mSystems submission. I agree with the previous editor that your study is scientifically sound and I am pleased to inform you that your manuscript has been editorially accepted for publication. However, there are a few additional questions in the submission form that need to be answered before the final decision. Once these are completed, please return your submission so that I can move your paper forward to acceptance.

Revision Guidelines

Sincerely,
Jan Claesen
Editor
Microbiology Spectrum

Re: Spectrum02559-24R1 (An exploratory study: the gut microbiome associates with immune checkpoint inhibitor response in individuals with non-melanoma skin cancers)

Dear Dr. Fyza Y. Shaikh:

Thanks for addressing the last few items in the Spectrum submission system. Your paper has now been accepted! Congrats!

Your manuscript has been accepted, and I am forwarding it to the ASM production staff for publication. Your paper will first be checked to make sure all elements meet the technical requirements. ASM staff will contact you if anything needs to be revised before copyediting and production can begin. Otherwise, you will be notified when your proofs are ready to be viewed.

Sincerely,
Jan Claesen
Editor
Microbiology Spectrum